# Bridging the condensation-collision size gap: a direct numerical simulation of continuous droplet growth in turbulent cloud

Sisi Chen[1], Man-Kong Yau[1], Peter Bartello[1], and Lulin Xue[2]

[1]McGill University, Montréal, Québec, Canada
[2]National Center for Atmospheric Research, Boulder, Colorado, USA

**Correspondence:** Sisi Chen (sisi.chen@mail.mcgill.ca)

**Abstract.** In most previous direct numerical simulation studies on droplet growth in turbulence, condensational growth and collisional growth were treated separately. Studies in recent decades have postulated that small-scale turbulence may accelerate droplet collisions when droplets are still small when condensational growth is effective. This implies that both processes should be considered simultaneously to unveil the full history of droplet growth and rain formation. This paper introduces the first DNS approach to explicitly study the continuous droplet growth by condensation and collisions inside an adiabatic ascending cloud parcel. Results from the condensation-only, collision-only, and condensation-collision experiments are compared to examine the contribution to the broadening of droplet size distribution by the individual process and by the combined processes. Simulations of different turbulent intensities are conducted to investigate the impact of turbulence on each process and on the condensation-induced collisions. The results show that the condensational process promotes the collisions in a turbulent environment and reduces the collisions when in still air, indicating a positive impact of condensation on turbulent collisions. This work suggests the necessity to include both processes simultaneously when studying droplet-turbulence interaction to quantify the turbulence effect on the evolution of cloud droplet spectrum and rain formation.

## 1 Introduction

Theoretical studies indicate that for droplets in the size range of 15-30$\mu m$ in radius, referred to as the condensation-collision size gap, neither condensational growth nor collisional growth is effective (Pruppacher and Klett, 1997) in producing precipitation. Classical parcel models generally yielded very narrow droplet size distributions (DSDs) and take a rather long time to form rain (Jonas, 1996). In nature, wide DSDs and large droplets are frequently observed in cumulus and even statocumulus clouds (e.g., Brenguier and Chaumat, 2001; Pawlowska et al., 2006; Prabha et al., 2012). This size gap problem represents a longstanding challenge in the ongoing quest to understand the warm-rain initiation process. In the literature, various mechanisms have been proposed to accelerate rain development, such as small-scale turbulence (Vaillancourt and Yau, 2000), the presence of giant aerosols (Johnson, 1982; Blyth et al., 2003; Jensen and Nugent, 2017), entrainment of unsaturated air (Baker et al., 1980; Lasher-trapp et al., 2005; Cooper et al., 2013) and large-eddy hopping (Cooper, 1989; Grabowski and Abade, 2017). This study focuses on the effect of small-scale turbulence containing eddies in the inertial and dissipation range with

length-scales $\ll$ 10 m as shown in Fig. 1 of Grabowski and Wang (2013) which can be resolved by the technique of direct numerical simulation (DNS).

Several mechanisms related to turbulence have been proposed to explain the fast growth of droplets in the condensation-collision size gap (Devenish et al., 2012; Grabowski and Wang, 2013). As a result of the response of droplet inertia to turbulent eddies of different scales, turbulent flow creates two effects: the non-uniform distribution of cloud droplets (clustering effect) and the increase in the relative velocities between droplets (transport effect). A number of DNS studies have reported that the geometric collision rate of droplets increases as turbulence intensifies (Franklin et al., 2005; Ayala et al., 2008; Onishi and Seifert, 2016). Concomitantly, turbulence modifies the response of a droplet to the local disturbance flow induced by other droplets through hydrodynamic interactions to increase the collision efficiency (Wang et al., 2008; Onishi et al., 2013; Chen et al., 2018). In particular, Chen et al. (2018) demonstrated that the turbulence enhancement of collisions became most significant among droplet pairs of similar sizes, suggesting that turbulence may efficiently broaden the narrow DSD generated from condensational growth.

Moreover, it has also been argued that the supersaturation perturbation field can arise from the fluctuation of temperature and water vapor in turbulence and the differential local water vapor consumption (Srivastava, 1989) which is enhanced by droplet clustering. This may lead to a distinct growth history by condensation for each droplet as it is transported in a turbulent flow (Lanotte et al., 2009). However, several DNS studies found that small-scale turbulence can only create small, if not insignificant, drop size broadening through condensation (Vaillancourt et al., 2002; Lanotte et al., 2009; Sardina et al., 2015). The reason is that the average time that droplets are exposed to supersaturation perturbations shortens as the turbulence intensifies and as droplets grow larger and begin to sediment (Vaillancourt et al., 2002). Lanotte et al. (2009) reported a wider size distribution when the Reynolds number of the flow, which was calculated based on the computational domain size, increased from 40 to 185 and proposed a simple scaling to extrapolate the DNS result to the typical size of a adiabatic cloud core (approximately 100 m wide or Reynolds number $\approx$ 5000). However, caution should be exercised in applying this scaling as DNS is not able to capture the spatiotemporal complexity of the turbulence at scales larger than the size of the domain. Sardina et al. (2015) also used a similar model as Lanotte et al. (2009) but extended the simulation time to 20 minutes to be comparable to the formation time of rain revealed in real observations. They found that the variance of the droplet size distribution was mainly determined by the large-scale flow, i.e., the large-hopping effect suggested by Grabowski and Wang (2013) and studied by Grabowski and Abade (2017). Nevertheless, it should be noted that their conclusion was based on the simplified assumption that both the mean updraft speed and the mean supersaturation were zero. On the other hand, the DNS model of Toshiyuki et al. (2016) considered a time-dependent and buoyancy-driven mean vertical motion calculated from a given environmental sounding. In their study of the effect of turbulence and entrainment on the evolution of cloud droplets, it was found that the thermodynamic fluctuations caused by turbulent advection prevented the buildup of the buoyancy force, leading to an even slower evolution of the mean droplet size and the vertical velocity as compared to those predicted by a parcel model.

A common limitation shared by most, if not all, previous DNS studies is that, the condensation process and collision-coalescence process were studied separately. This assumption may be justifiable in a parcel model due to the non-overlapping droplet-size regimes of the two growth processes in still-air. However this assumption is questionable in DNS studies which

reveal substantial turbulent enhancement of collisions among droplets in the condensation-collision size gap. As there is an absence of DNS work on continuous droplet growth incorporating both processes, it is the goal of this study to unveil the full history of droplet growth and the DSD broadening by condensational and collisional growth in a turbulent, supersaturated environment undergoing an adiabatic ascent.

The purpose of this study is: 1) to introduce the first DNS approach to explicitly resolve the continuous droplet growth by condensation and collision in shallow, turbulent clouds, and 2) to answer the following two questions: "how does the droplet collisional process interact with the droplet condensational growth process?" and "what is the role of turbulence in this interaction? "

Our approach is to incorporate the droplet hydrodynamic collision and condensation processes into a single DNS modeling framework. Arguably, this model provides a first direct approach to bridge the condensation-collision gap that has puzzled the cloud physics community for decades. The paper is organized as follows. In Sect. 2 we describe the sets of equations adopted from Vaillancourt et al. (2001) and Chen et al. (2018) and the accompanying modification. The simulated results from three sets of experiments (condensation-only, collision-only, condensation-collision) in various turbulent environments are given in Sect. 3, to be followed by a conclusion and remarks on the limitation of this study in Sect. 4.

## 2   Model description and experimental setup

This paper represents a sequel to Chen et al. (2018) as part of our on-going exploration of the evolution of cloud DSD affected by turbulence. The DNS model adopted was originally developed by Vaillancourt et al. (2001) in perhaps one of the earliest DNS approaches to simulate droplet growth in turbulence. Vaillancourt et al. (2001) focused on the impact of turbulence on droplet condensation, and thus collisions were not considered. A number of extensions followed. Franklin et al. (2005) resolved the droplet collisions using an efficient collision detection technique. Chen et al. (2016) made changes to allow simulation in larger domain sizes and introduced a new forcing scheme to achieve a statistically steady turbulent dissipation rate. Chen et al. (2018) added the local disturbance flow field induced by droplets to obtain accurate turbulent collision efficiencies and droplet collisional growth affected by both the disturbance flow and the turbulence flow.

In the present study, the model from Chen et al. (2018) is further extended to restore the thermodynamical framework of Vaillancourt et al. (2001) to include condensational growth. Specifically, the whole DNS box is regarded as a parcel ascending adiabatically from near the cloud base with a constant mean updraft. Two sets of equations are used to solve for 1) the macroscopic variables that describe the time evolution of the parcel mean state properties and 2) the microscopic variables that describe the turbulent flow, as well as the temperature and the water vapor mixing ratio fluctuation fields. Furthermore, equations pertaining to the thermodynamics are modified to improve the accuracy of droplet condensational growth. For convenience of reference, a list of constants is given in Appendix A and the detailed equations are provided in Appendix B.

In the presence of the thermodynamic fluctuation fields and the turbulence flow field, droplets grow in two distinct ways simultaneously:

1) Droplets grow by condensation with its growth rate directly proportional to the instantaneous supersaturation (see Eq. (B1) in Appendix B). When a droplet moves relative to the air, the water vapor field is not spherically symmetric around the droplet surface but is modified depending on the direction of motion (so-called the ventilation effect). This effect becomes important when droplets are greater than 30 $\mu m$ in radius (Sedunov, 1974). In Vaillancourt et al. (2001), all droplets were smaller than 20 $\mu m$ and this effect was not considered. However, the present study allows droplets to grow larger and thus the ventilation coefficient is added to the droplet growth equation. Following Vaillancourt et al. (2001), the curvature term and the solute term are neglected in the equation, and the droplets are treated as pure water drops since all droplets in this study are greater than 5 $\mu m$ (Pruppacher and Klett, 1997).

2) Simultaneously, droplets grow through the collision-coalescence process. The droplet motion and collisional growth are treated in the same manner as in Chen et al. (2018). Each droplet is tracked in the Lagrangian framework, with its motion determined by gravity and the local fluid drag force (Eq.(1) in Chen et al., 2018). Once two droplets collide, they coalesce to become a bigger entity with its mass equal to the sum of the masses of the collided droplets and its location being the barycenter of the binary system before the collision. The velocity of the coalesced droplet is calculated based on the conservation of momentum. Since we are particularly interested in the condensation-collision size range, i.e., droplets smaller than drizzle drops, defined as drops with a radius equal to larger than 100 $\mu m$, our study only consider radius $r \ll 100 \mu m$. In addition, solving the motion of large drops requires more complex consideration such as induced turbulent wakes and drop deformation which are beyond the scope of this study. Therefore, droplets reaching 100 $\mu m$ are considered as fall-outs and are not allowed to grow further, i.e., they neither interact with other droplets nor affect the local disturbance flows. It should be noted that this assumption bears certain caveats. The Stokes' Law assumption for the disturbance flow becomes less accurate for droplets larger than 50 $\mu m$, because droplets over 50 $\mu m$ (and smaller than 100 $\mu m$) have a particle Reynolds number of order one. However, since the collision efficiency for droplets larger than 50 $\mu m$ is close to unity due to the large Stokes number, it is argued that the impact of the disturbance flow on the collision statistics of those large particles would be secondary. Furthermore, in all the simulations the calculated total number of collisions remains below 10% of the total number of droplets. Specifically, the number of collisions is within 9% in strong turbulence and below 3% in weak turbulence and below 2% in still air. It follows that the impact of reducing the droplet number concentration due to collisions on the resulting DSD can be assumed small. One alternative post-collision treatment maybe to introduce a new, randomly located droplet into the domain once a collision happens so that the droplet number concentration remains constant. However, the size of droplets that should be introduced remains contentious and needs further justification.

Three sets of experiments are conducted to evaluate the DSD broadening due to the turbulence effect on different droplet growth processes: 1) droplet growth by condensation only (referred to as the condensation-only experiment), 2) droplet growth by collision-coalescence only (referred to as the collision-only experiment), and 3) droplet growth by condensation and collision-coalescence together (referred to as the condensation-collision experiment), respectively. All experiments use the same initial DSD shape adopted from an aircraft measurement in non-precipitating cumulus clouds (Raga et al., 1990). The initial droplet number concentration is set as 80 $cm^{-3}$ and a constant updraft of 2.5 $ms^{-1}$ is used to represent the condition of pristine maritime cumulus clouds.

For each set of experiments (except for the condensation-only experiment), three flow configurations are considered: purely-gravitational case (i.e., still air), a weak turbulence case (with eddy dissipation rate $\epsilon = 50\ cm^2 s^{-3}$), and a strong turbulence case (with $\epsilon = 500\ cm^2 s^{-3}$). The domain size of each simulation is about 10 cm in each direction, with grid space $\approx 0.1$ cm determined by the dissipation rate as explained in Chen et al. (2016). It is recognized that droplet condensation in still-air leads to a narrow DSD and the DSD broadening by condensation impacted by small-scale turbulence is insignificant. Therefore, during the condensation-only experiment, only the strong turbulence simulation is performed to serve as an upper bound of the DSD broadening among the three flow conditions. As a result, seven simulations are performed. Each simulation lasts 6.5 minutes of real-time which is the approximate duration for the whole parcel to ascend from cloud base to 1000 m above the base, representative of a typical cumulus development.

## 3  Results and discussion

We first compare the results from the three experiments to scrutinize the contributions of the different droplet growth processes under the effect of turbulence. Figure 1 shows the DSDs at the end of each experiment in strong turbulence. As a reference, the initial DSD is displayed with a gray area. It should be noted that droplet number concentrations below 0.001 $cm^{-3}$ will be treated as statistical uncertainty throughout the discussion, since they correspond to less than 2-4 droplets in the domain. Consistent with past findings, the turbulence effect on droplet condensational growth is small. The condensation-only process produces the narrowest size distribution among the three experiments and droplets grow no larger than 20 $\mu m$ at the end of the simulation. On the other hand, in both the collision-only experiment (blue curve) and the condensation-collision experiment (yellow curve), a substantial number of large droplets are found. Furthermore, compared to the collision-only simulation, the condensation-collision experiment generates more large droplets and substantially larger droplets. The largest $r$ reaches 100 $\mu m$ at the end of the simulation compared to less than 65 $\mu m$ in the collision-only case. Meanwhile, the number concentration of $r > 30\ \mu m$ droplets in the condensation-collision case increases by a factor of 2.3 (0.35 $cm^{-3}$ compared to 0.15 $cm^{-3}$ in the collision-only case).

To examine whether this enhanced broadening due to the inclusion of the condenSational process depends on the flow, detailed comparisons between the collision-only and the condensation-collision experiments are made under three flow conditions: purely-gravitational, weak turbulence, and strong turbulence. Figure 2 demonstrates the time evolution of the DSD in the two sets of experiments under the three different flows. It is found that:

1) In the purely-gravitational case (Figs. 2(a) and 2(b)), despite the condensation-collision experiment producing larger maximum droplets at the end of the simulation relative to the collision-only experiment (black outline in Fig. 2), the number concentration of large droplets is still negligible (as $r > 35\ \mu m$ droplets stay below 0.001 $cm^{-3}$ as seen from the expansion of the purple edge with time);

2) In the turbulent cases, we find more large droplets and much larger maximum droplet sizes in the domain when condensational growth is considered. With weak turbulence droplets larger than 35 $\mu m$ (over 0.001 $cm^{-3}$) can be seen as early as 3.5 minutes in the condensation-collision experiment, but 6 minutes in the collision-only run. With strong turbulence large droplets

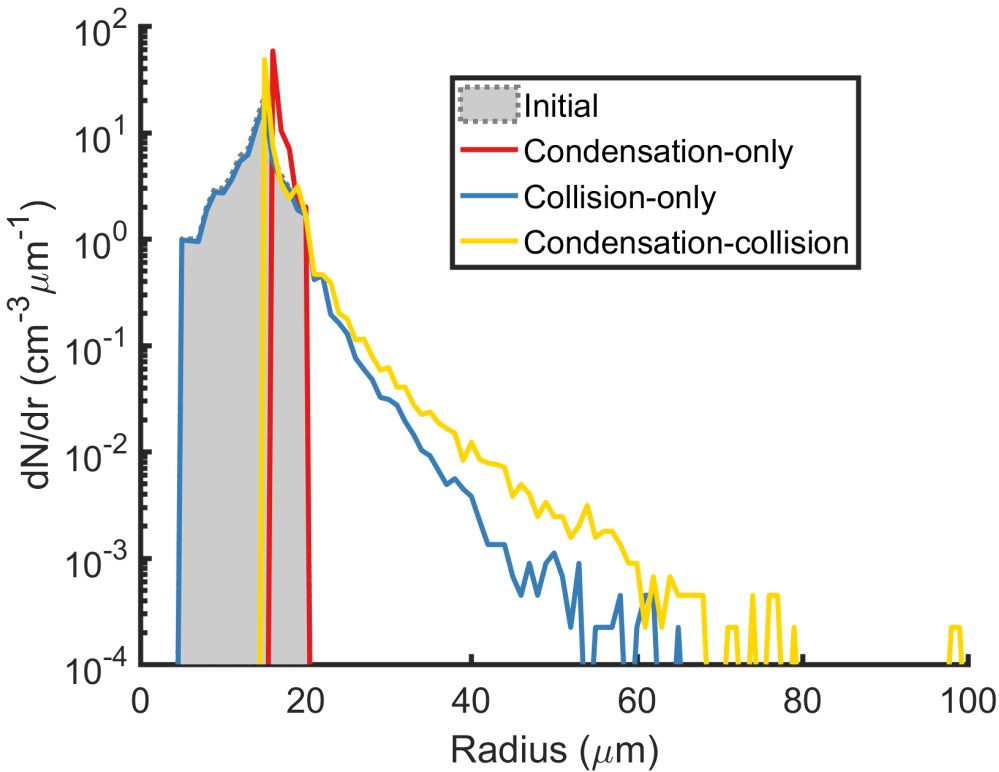

**Figure 1.** Droplet size distributions at the 6.5th minute for the condensation-only case (red), collision only case (blue) and condensation-collision case (yellow). Dissipation rate is 500 $cm^2 s^{-3}$ for all cases with the initial size distribution shown as a dashed grey line. Droplet number concentrations below 0.001 $cm^{-3}$ are treated as statistical uncertainty.

were found in the 3rd minute in the condensation-collision simulation compared to the 4th minute without condensation. It is evidence that both experiments experience earlier formation of large droplets as turbulence intensifies while the inclusion of condensation further accelerate the droplet growth. This result evinces that an effective condensation-collision broadening mechanism exists that strengthened with increasing turbulence intensity.

5     A condensation-induced broadening has been found in all three flow conditions, though it seems that it is negligible in the case of still air. This phenomenon can be explained by two main mechanisms:

    1) The condensational growth process effectively produces droplets of small sizes ($r < 10$ $\mu m$) to medium size ($10-20$ $\mu m$) due to the fast growth rate of small droplets. This conjecture is supported by the result on the right column of Fig. 2 showing that among the three condensational cases, all droplets smaller than 15 $\mu m$ become greater than 15 $\mu m$ within 4 minutes. As

10   bigger droplets have higher collision rates, the average collision rate in the domain is expected to increase progressively as more medium-sized droplets are formed through condensation, and they become more likely to be collected by other droplets.

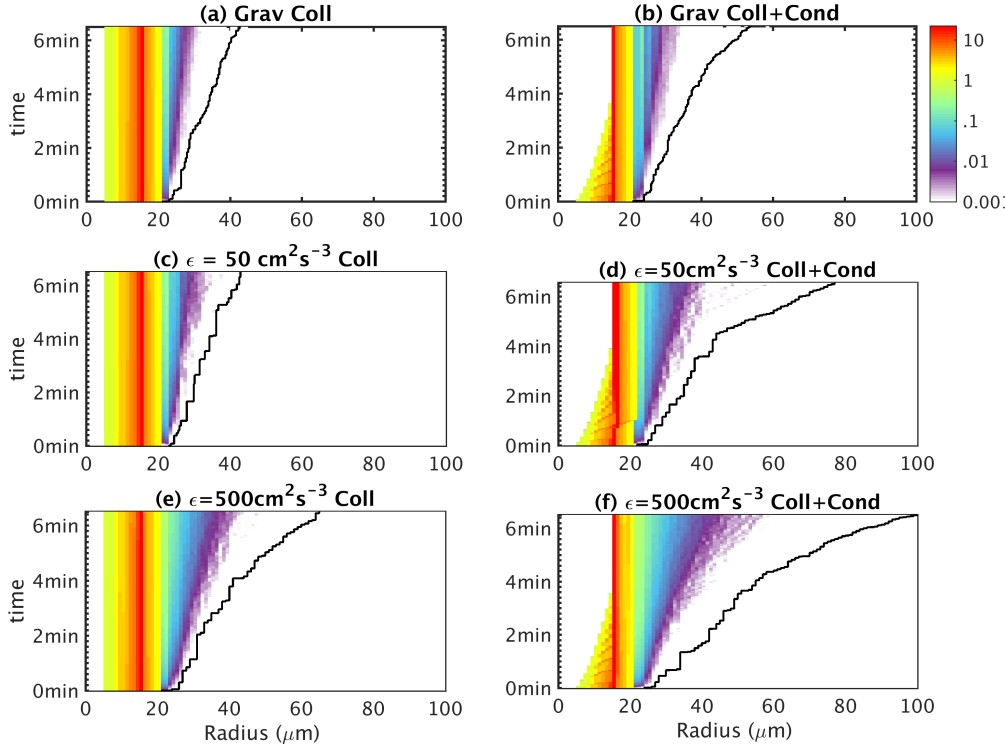

**Figure 2.** The time evolution of the DSD in the collision-only experiments (left column) and the collision-condensation experiments (right column). Results from the purely-gravitational case (first row), weak turbulence ($\epsilon = 50\ cm^2s^{-3}$, second row), and strong turbulence ($\epsilon = 500\ cm^2s^{-3}$, third row) are demonstrated. The solid black curve indicates the largest droplet of the entire domain. The droplet number concentration ($cm^{-3}$) on each size bin (bin width = 1 $\mu m$) is displayed in color using a logarithmic scaling shown in the color bar. Droplet number concentrations below 0.001 $cm^{-3}$ are treated as statistical uncertainty and thus are given no color in the plot.

2) Condensational growth narrows the DSD and providing a great number of similar-sized droplets (i.e., the radius ratio between the small droplet and large droplet, r/R, is close to unity). Chen et al. (2018) found that turbulence enhancement of collision rate is most significant in similar-sized droplets, and stays relatively weak for $0.2 < r/R < 0.8$ (Fig. 3 in their paper). In an environment with similar-sized droplets created by condensation, the turbulence-enhanced collisions are enhanced to accelerate the production of large droplets.

The first mechanism of enhanced collision rate due to larger mean droplet sizes can also happen in the purely-gravitational case but will be offset by the inefficient gravitational collection process due to the DSD narrowing by condensation. In turbulent cases, the condensational DSD produce similar-sized droplets to allow the turbulence-enhanced similar-sized collision process to act, leading to a positive feedback mechanism. Evidence for this hypothesis can be found by comparing the probability distribution function (PDF) of collisions with respect to r/R (the radius ratio between the small droplet and big droplet in

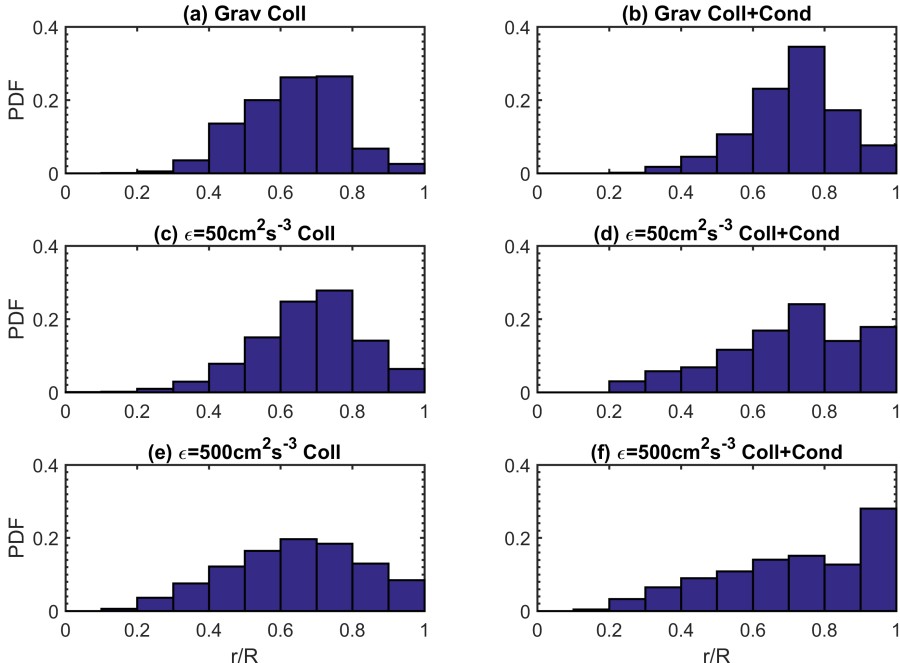

**Figure 3.** Probability distribution function (PDF) of collisions with respect to $r/R$ at three different flow conditions (first row: pure gravity, second row: weak turbulence, third row: strong turbulence). Results from the collision-only experiments (the left column) and the condensation-collision experiments (the right column) are shown for comparison

a droplet pair) in the collision-only and the collision-condensation experiments. As seen in Fig. 3, the PDF of collisions in either the weak turbulence or the strong turbulence become more flattened when condensation is included. In particular, the chance of similar-sized collisions ($r/R > 0.9$) is substantially greater. On the contrary, a narrower PDF is found in the purely-gravitational case (Figs. 3(a) and 3(b)). Figure 4 demonstrates the distributions of collision frequency and the collision

5  enhancement due to condensation. It is found that in the purely-gravitational case the number of similar-sized collisions doubles in the condensation-collision experiment, which results from the increased number of similar-sized droplets introduced by condensation. It is obvious that increasing the intensity of turbulence further enhances these collisions. The similar-sized collisions increase by a factor of 3.5 in weak turbulence and a factor of 4.5 in strong turbulence.

In the small r-ratio range ($r/R < 0.7$), the total collisions in the purely-gravitational case are lowered by more than half due to

10  a reduced number of those droplet pairs (Fig. 4(g)). However, in the turbulent cases, the collision frequency instead experiences a mild increase compared to the collision-only experiment. This increase is due to the fact that condensation increases the population of medium-sized droplets ($r = 10 - 20\mu m$) and turbulence continues to enhance the collisions of these droplets. The abundant number of those medium-sized droplets boosts the number of similar-sized collisions by turblence to produce larger droplets. Meanwhile, the larger size from growth by condensation substantially increases the chance of those droplets to

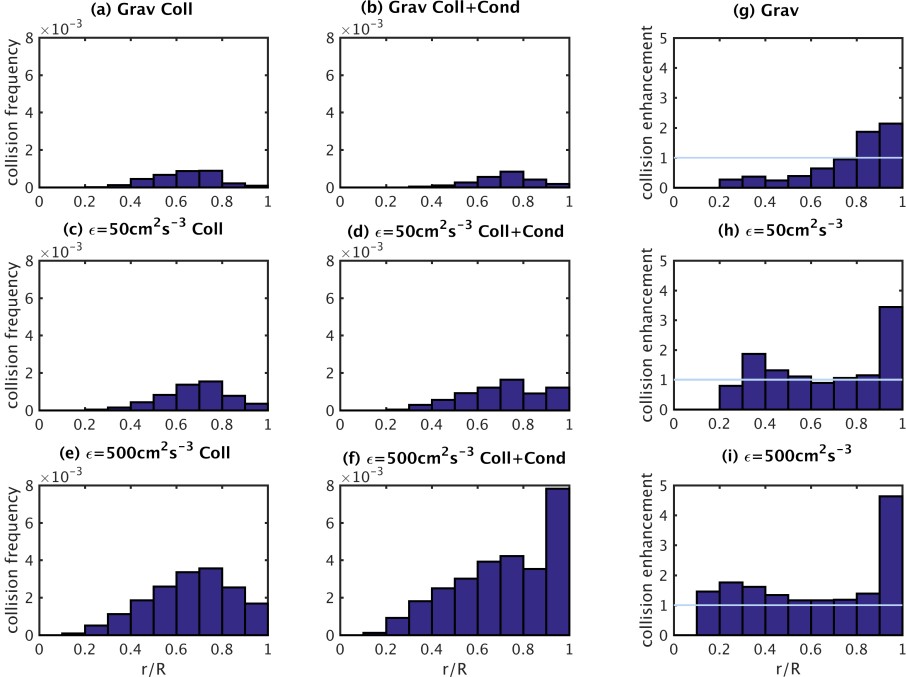

**Figure 4.** Subplot (a)-(f) are the same as in Fig. 3 but for the collision frequency $(cm^{-3}s^{-1})$ from different r/R pairs. Subplot (g)-(i) are the enhancement of collision frequency for different r/R pairs due to the inclusion of condensation. The enhancement is calculated by taking the ratio of the collision frequency from the condensation-collision experiment and the collision-only experiment. Results from three different flow conditions are demonstrated.

be collected by other larger droplets. Furthermore, the formation of large droplets due to the turbulence-enhanced collisions in turn contributes to the growing collector droplet population, thus further increasing the chance of these medium-sized droplets to be collected. In the purely gravitational case, this process is inhibited by the insignificant similar-sized collisions in spite of their number being doubled from the collisional-only case.

5      To further illustrate the influence of turbulence on the enhancement of the condensation-induced collisions, we have studied the impact of condensation on the evolution of the number of droplet pairs. We divided the droplet pairs into two groups: the similar-sized pair group with $r/R > 0.7$ and the different-sized pair group with $r/R \leq 0.7$. We then calculated the total number of droplet pairs within the two groups. By comparing the results from the collision-only experiments and the collision-condensaction experiments, we are able to separate the enhancement of collision rate solely due to turbulence and the
10   enhancement directly associated with the inclusion of condensational growth.

     Fig. 5 shows the time evolution of the total number of droplet pairs for the two groups in the domain. For the convenience of comparison among the three turbulent cases, the pair numbers are calculated based on the droplet number concentration $(cm^{-3})$. For example, for the droplet pair of $r_1$ and $r_2$ with concentrations of $n_{d1}$ and $n_{d2}$, the pair number is $n_{d1}n_{d2}$ if $r_1 \neq r_2$

and $\frac{n_{d1}n_{d2}}{2}$ if $r_1 = r_2$. Therefore the unit of the pair number is in $cm^{-6}$. It is found that in the collision-only experiment the number of different-sized droplet pairs stays relatively constant (Fig. 5(a)). Meanwhile, the number of similar-sized droplets undergoes only a weak decay (Fig. 5(c)). Compared to the pure gravity case, turbulence effectively accelerates similar-sized collisions, while the enhancement of different-sized collisions is relatively small. On the one hand, the turbulence enhancement of similar-sized collisions is due to the fact that turbulence has a stronger effect on the similar-sized collision efficiency (Chen et al., 2018). On the other hand, it has been found that the turbulence clustering effects are more significant for droplets of similar sizes (Chen et al., 2016). They tend to cluster in the same regions of the flow because of similar droplet inertia and terminal velocities. This effect has been confirmed previously in a number of studies (e.g., Ayala et al., 2008; Franklin et al., 2005) and is especially pronounced for large droplets. The reason is that small droplets have small Stokes numbers, and they adjust very quickly to changes in the flow and therefore behave more like fluid tracers than inertial droplets. Consequently, with growing droplet size, turbulence clustering of similar-sized droplets becomes more significant and the number of similar-sized pairs undergo an accelerated decline. This can be seen on Fig. 5(c) where the curve of the turbulent case deviates from the pure-gravitational case.

By contrast, in the condensation-collision experiment the trend of the number of droplet pairs behaves in a more complex fashion due to the inclusion of condensation. As illustrated by Fig. 5(b) and (d), the droplet growth experiences two different stages. The number of different-sized pairs significantly decreases in the first two minutes mainly due to the rapid condensational growth of droplets with $r < 15\mu m$. This is demonstrated in Fig 2 where the droplet number concentration for $r < 15\mu m$ quickly reduces from larger than $1 cm^{-3}$ to below $0.001 cm^{-3}$ in the first two minutes, while the production of large droplets is still negligible. Concurrently, the number of similar-sized droplets significantly increases during the first two minutes and steadily decreases thereafter (Fig. 5(d)). The large increase of similar-sized pairs in the collision-condensation experiments during the first two minutes significantly increases the number of turbulent-enhanced similar-sized collisions. After two minutes, the condensational effect diminishes and the collision-coalescence process takes over in modulating the droplet pair population. The subsequent decline of the number of similar-sized pairs and the increase in the number of the different-sized pairs mainly arise from the collision-coalescence process.

## 4 Conclusions

This work provides the first DNS study to explicitly resolve continuous droplet growth by condensation and collision in a turbulent environment. The results are expected to contribute toward resolving the warm-rain initiation problem.

Results from the condensation-only, collision-only, and condensation-collision experiments are compared to examine the contribution to the DSD broadening by the individual process and by the combined processes acting in concert. Three different flow environments (still air, weak turbulence, and strong turbulence) are investigated to scrutinize the impact of turbulence in the condensational-induced collisions. By comparing the collision frequencies of the collision-only experiment and the condensation-collision experiment, it is found that condensational growth boosts the collisions when the flow is turbulent and slows down the collisions for the case of still air.

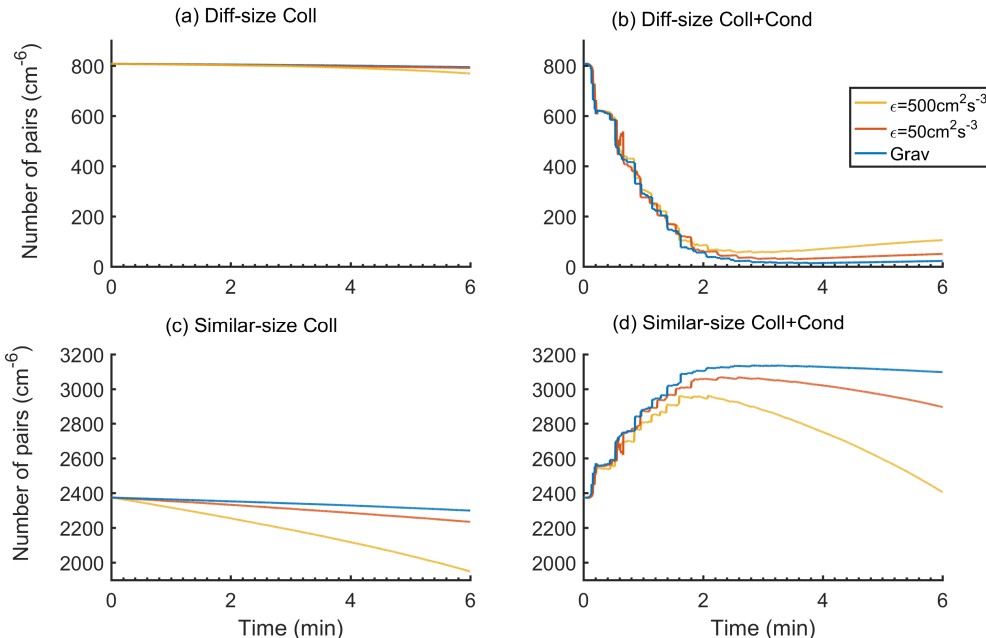

**Figure 5.** Time evolution of the number of pair combinations for (a) the different-sized droplets ($r/R \leq 0.7$) and (c) the similar-sized droplets ($r/R > 0.7$) in the collision-only experiments, and (b) the different-size droplets and (d) the similar-sized droplets in the condensation-collision experiments. The pair combination is computed using the droplet number concentration ($cm^{-3}$), therefore the unit is in $cm^{-6}$. The color denotes the three different flow conditions which are shown in the legend.

In the purely-gravitational experiment, the abundant similar-sized droplets generated by condensation inhibit the gravitational collection process, and the collision frequency of $r/R < 0.7$ reduces by half. As a result, the number concentration of droplet larger than 35 $\mu m$ remains lower than 0.001 $cm^{-3}$ throughout the simulation.

In the turbulence experiments, a greater number of large droplets are produced, and their appearance occurs faster as turbulence intensifies, implying an effective turbulence impact on droplet size broadening. Furthermore, droplets larger than 35 $\mu m$ form 1-2 minutes earlier in the collision-condensation experiments. It follows that these droplets appear as early as the 3rd minute in strong turbulence situation. This result suggests that a sophisticated model that takes into account both the turbulence-enhanced collisions and condensational-induced collisions under the effect of turbulence should be used to study the cloud droplet spectrum broadening and rain formation.

Finally we remark on the limitation of this study and some suggestions for future work. It has been found that the evolution of the DSD and the rain formation time highly depend on the initial shape of the DSD and the droplet number concentration. Therefore, simulations of different initial DSDs are to be conducted to better understand its dependency. In addition, the initial DSD used in this study is taken from flight observations which represents an average over a long sampling time and a wide sampling volume. In this case, the initial DSD is not guaranteed to be representative of the steady-state DSD from

aerosol activation and condensational growth in adiabatic cloud cores. However, with the continuous advancement of in-situ and laboratory measurement technology such as HOLO-DEC (Glienke et al., 2017) and PI chamber (Desai et al., 2018), representative sampling of the DSD near the cloud base inside adiabatic cores may be possible in the near future. It is also desirable to include the aerosol activation process to enable cloud particles to grow from the very beginning (i.e., dry aerosols in sub-cloud regions). We strive to explore this approach in a future study. Besides, the model can also be modified to study other microphysics processes such as ice nucleation which is poorly parameterized for deep convective clouds and cirrus clouds, and particle electrification which is potentially important in aerosol scavenging and droplet collisions.

## Appendix A: List of constants

See list of constants in Table A1

**Table A1.** List of constants

| | |
|---|---|
| $D_v = 2.55 \times 10^{-5}$ | water vapor diffusivity $[m^2 s^{-1}]$ |
| $D_t = 2.22 \times 10^{-5}$ | thermal diffusivity $[m^2 s^{-1}]$ |
| $\nu = 1.6 \times 10^{-5}$ | air kinematic viscosity $[m^2 s^{-1}]$ |
| $K_a = 2.48 \times 10^{-2}$ | thermal conductivity of air $[Jm^{-1}s{-1}K^{-1}]$ |
| $R_v = 461.5$ | Individual gas constant for water vapour $[JKg^{-1}K^{-1}]$ |
| $R_a = 287$ | Individual gas constant for dry air $[JKg^{-1}K^{-1}]$ |
| $L = 2.477 \times 10^6$ | specific latent heat for water $[JKg^{-1}K^{-1}]$ |
| $C_p = 1005$ | specific heat for air $[JKg^{-1}K^{-1}]$ |
| $\Gamma_d = -g/C_p$ | dry adiabatic lapse rate |
| $\rho_w = 1000.0$ | density of water $[Kgm^{-3}]$ |
| $S_{ch} = \nu/D_v$ | Schmidt number |

## Appendix B: Equations for DNS model

### B1  Microscopic equations

The condensational growth rate of an individual droplet with radius $R_i$ is as follow:

$$\frac{dR_i^2}{dt} = 2K f_v S. \tag{B1}$$

Here $K^{-1} = \frac{\rho_w R_v T}{e_{sat}(T) D_v} + \frac{L\rho_w}{K_a T}\left(\frac{L}{R_v T} - 1\right)$, where $e_{sat}$ is the saturated water vapor pressure. $f_v$ refers to the droplet ventilation coefficient. The value of $f_v$ is determined by the empirical formulas from laboratory experiment of Beard and Pruppacher (1971):

$$f_v = 1.0 + 0.108(N_{Sc}^{1/3} Re_p^{1/2})^2, \text{ for } N_{Sc}^{1/3} Re_p^{1/2} < 1.4, \tag{B2}$$

$$f_v = 0.78 + 0.308(N_{Sc}^{1/3} Re_p^{1/2}), \text{ for } 51.4 > N_{Sc}^{1/3} Re_p^{1/2} \geq 1.4, \tag{B3}$$

where $Re_p = \frac{2R_i|\mathbf{V}|}{\nu}$ is the droplet Reynolds number, $\mathbf{V}$ is the velocity of droplet i. $S$ is the supersaturation in the grid cell where droplet i is located, defined as

$$S = \frac{q_v}{q_{vs}} - 1, \tag{B4}$$

where $q_v$ is the water vapor mixing ratio, with its corresponding saturated value $q_{vs}$ determined by temperature ((2.17)-(2.18) in Rogers and Yau 1989). We assume that all droplets residing in the same grid cell are exposed to the same supersaturation environment. The scaler fields of $q_v$ and temperature $T$ can be decomposed into the parcel mean state and the perturbation state. The parcel mean state is calculated via the macroscopic set of equations shown in Sect. B2 and the perturbations are calculated as follows:

$$\frac{\partial T'}{\partial t} = -\nabla \cdot (\mathbf{U}T') - W'\Gamma_d + \frac{L}{C_p}C_d' + D_t\nabla^2 T', \tag{B5}$$

$$\frac{\partial q_v'}{\partial t} = -\nabla \cdot (\mathbf{U}q_v') - C_d' + D_v\nabla^2 q_v', \tag{B6}$$

where $W'$ is the vertical perturbation velocity. $C_d' = C_d - C_{dM}$ is the differential condensation rate between the grid cell and the whole parcel. Given (B1), the condensation rate inside the grid cell can be simplified as:

$$C_d = \frac{1}{m_a}\sum_i^n \frac{4}{3}\pi\rho_w\frac{dR_i^3}{dt} = \frac{4}{m_a}\pi\rho_w K f_v \sum_i^n R_i S. \tag{B7}$$

The turbulent velocity field $\mathbf{U}$ is governed by the incompressible Navier-Stokes equations:

$$\frac{\partial \mathbf{U}}{\partial t} + (\mathbf{U}\cdot\nabla)\mathbf{U} = -\frac{1}{\rho_a}\nabla P + \nu\nabla^2\mathbf{U} + \mathbf{F}, \tag{B8}$$

$$\nabla \cdot \mathbf{U} = 0, \tag{B9}$$

where P is the perturbation pressure deviation from the hydrostatic pressure $P_M$. The pressure term can be dropped when the equations are solved in vorticity form. $\mathbf{F}$ is the external forcing. We used the forcing method of Chen et al. (2016) to maintain the turbulence. The droplet motion is governed by fluid drag force and gravity:

$$\frac{d\mathbf{V}(t)}{dt} = \frac{\mathbf{V}(t) - \tilde{\mathbf{U}}(\mathbf{X}(t),t)}{\tau_p} + \mathbf{g}, \tag{B10}$$

$\tau_p$ denotes the droplet response time. For $r < 40\mu m$, Stokes drag force is applied and $\tau_p = (\frac{2\rho_w}{9\nu\rho_a})r^2$. Droplet terminal velocity can be obtained using $V_T = g\tau_p$. For $r \geq 40\mu m$, the terminal velocity derived from the experimental data is applied to those big droplets: $V_T = k_2 r$, here $k_2 = 8 \times 10^3 s^{-1}$ (Rogers and Yau, 1989, p.126). $\tilde{\mathbf{U}}$ is the flow velocity at the droplet center, contributed by the turbulent flow field $\mathbf{U}$ and the disturbance flow $\mathbf{U_{dist}}$ caused by neighboring droplets (Chen et al., 2018). The superposition method by Wang et al. (2005) is used to calculate the disturbance flow.

## B2 Macroscopic equations

The time evolution of the parcel-mean temperature $T_M$, water vapor mixing ratio $q_{vM}$, pressure $P_M$, density $\rho_{aM}$ are described as below. All variables of parcel mean are denoted with a subscript M.

$$\frac{dT_M}{dt} = -W_M \Gamma_d + \frac{L}{C_p} C_d M \tag{B11}$$

$$\frac{dq_{vM}}{dt} = -C_{dM} \tag{B12}$$

$$C_{dM} = \frac{1}{M_a} \sum_i^N \frac{d}{dt} \left( \frac{4}{3} \pi \rho_w R_i^3 \right) = \frac{4}{M_a} \pi \rho_w K f_v \sum_i^N R_i S, \tag{B13}$$

$$\frac{P_M}{dt} = -\rho_{aM} g W_M \tag{B14}$$

$$\rho_{aM} = \frac{P_M}{R_a T_M} \tag{B15}$$

The total fields of T and $q_v$ are calculated by adding the macroscopic variables and the perturbation variables.

*Competing interests.* The authors declare that they have no conflict of interest.

15 *Acknowledgements.* We would like to acknowledge Dr. Paul A. Vaillancourt from Environment and Climate Change Canada for providing the original DNS code and offering constant help in this project. We also thank Dr. Jorgen Jensen and Dr. Hugh Morrison from NCAR for their valuable discussions. Special thanks to Dr. Yeti Li for his insightful comments in shaping the first draft of this paper. We would also like to thank Dr. Brian Dobbins and Dr. Jeremy Sauer from NCAR for their generous help in improving the code performance. L. Xue appreciates the support of Beijing Weather Modification Office through Beijing Municipal Science and Technology Commission (Grant
20 No.D171100000717001). We also acknowledge the support of the Natural Sciences and Engineering Research Council of Canada (NSERC). Computations were made on supercomputer Cheyenne (doi:10.5065/D6RX99HX) provided by NCAR's Computational and Information Systems Laboratory, sponsored by the National Science Foundation and on supercomputer Cedar provided by WestGrid (www.westgrid.ca) and Compute Canada Calcul Canada (www.computecanada.ca).

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
