# Peer review of "Bridging the condensation-collision size gap: a direct numerical simulation of continuous droplet growth in turbulent cloud"

_Atmospheric Chemistry and Physics, 2018_

## Referee Comment (RC1) · W. Grabowski (Referee) · 8 Mar 2018

Review of "Bridging the condensation-collision size gap: a direct numerical simulation of continuous droplet growth in turbulent cloud" by Chen et al.

This paper reports a very nice study with some surprising results that can be published after relatively minor revisions. I have a couple more serious comments and a few technical points. All these should be straightforward to address with relatively small changes to the text. Overall, the writing is clear and the subject is suitable for the ACP journal.

[Figure]

General comments:

1. I found it surprising that the impact of turbulent collisions between similar-sized droplets overwhelms the gravitational collisions between droplets with different sizes. I do not say I do not believe the effect, I just think this should be emphasized more in the manuscript and perhaps supported by additional arguments. Yes, I agree that turbulence dominates the enhancement for equally-sized droplets simply because gravitational collisions vanish in that limit. This is the major problem when showing the turbulent enhancement. However, it is not clear to me why turbulent collisions between equally-sized droplets should outnumber gravitational collisions between droplets of different sizes. The argument presented in the paper, that is, narrowing of the droplet spectra through condensational growth leading to more collisions when turbulence is added, hinges on this conjecture. I think this is the crux of the argument and it should be appropriately stressed in the manuscript. Moreover, one may ask why it is so? Is this because droplets of different sizes tend to cluster in different regions of the turbulent flow (as shown by Lain-Ping Wang in some of his papers), but droplets of the same size should cluster in the same region? Is this the collision efficiency effect? I think it would be appropriate to expand the analysis and maybe come-up with a hand-waving argument to provide some additional support for the key argument.

2. I am little concerned with a small size of the computational domain used in the simulations. The size limits the range of scales that the simulations can cover, but this is not what I am worried about. There are some suggestions in the literature that claim the problem also depends on the Reynolds number, that is, the size of the domain, but I feel this is of secondary importance. I am worried about the number of droplets that simulations include. Assuming the total concentration is somewhere around 100 per cc, then you carry around about 105 droplets in your 1 liter domain. To create one 100 micron droplet out of a cloud of droplets with mean size of, say 10 microns, takes of the order of 1000 collisions. Thus, the number of droplets you carry has to significantly decrease with time. Is this a problem? Of course, this also means that

you underestimate the impact, correct? I feel one should discuss this issue in the paper (e.g., show how the number of droplet changes with time) and suggest some improvements. One is to run ensemble of simulations to provide confidence intervals on Fig. 1 (the oscillations for radius larger than about 40 microns are a result of a single realization, correct?). The other possibility is to add droplets, for instance, create a new droplet, but keep the colliding droplets in the domain (perhaps re-positioning them randomly). The support for such an approach may come from the following argument: larger droplets fall faster and they simply fall out from the volume you consider and find themselves in the environment that has the same droplet population as before the collision. Such a methodology would provide an upper bound of the impact, correct? I feel it would worthwhile to discuss this aspect in the paper.

Minor specific comments (P – page, L – line):

1. The text uses the word "observation" and "observe" in several places. I suggest to replace with different words to avoid confusion. This is a numerical not observational study.

2. P2, L26: What you mean by the "large-scale flow" here? Is that the eddy-hopping idea as suggested by Grabowski and Wang (the ARFM review) and studied in Grabowski and Abade (JAS 2017)?

3. P4, L15: The 100 micron drop is an arbitrary choice. You can argue it comes from the traditional separation between cloud droplets and drizzle drops, correct? Arguably, one should select a smaller size because the Stokes flow solution that you apply in hydrodynamic interaction calculations is only valid for droplets up to about 50 microns in radius. A comment on that would be desirable here.

4. P. 10, L15. I think there should be R2 on lhs of Eq. B1, correct?

Signed: W. Grabowski

---

## Referee Comment (RC2) · Anonymous Referee #2 · 13 Mar 2018

Review of
*Bridging the condensation-collision size gap: a direct numerical simulation of continuous droplet growth in turbulent cloud*
submitted to ACP by Chen, Yau, Bartello and Xue

SUMMARY: This manuscript discusses results from Direct Numerical Simulations at low resolution ($N^3 = 64^3$) of turbulent cloud environment: the flow is seeded with point-like droplets, and both condensational and collisional growth are studied. The authors consider three flow situations: 1) droplets settling in still air, ii) droplets moving in a flow characterised by a low value of the kinetic energy dissipation rate $\epsilon_0$, and iii) droplets moving in a flow characterised by a higher value of the turbulent kinetic energy dissipation rate $10\,\epsilon_0$. To quantify the effects of turbulence on the droplet growth, the size distribution (SD) obtained in the different runs (7 in total) is examined and compared.
The main finding of the paper is to show that, starting from the same initial condition for the droplets, the SD exhibits a larger broadening when both condensational and collisional effects are implemented (see below).
In a previous paper (Ref.[1]), the authors used DNS to study turbulence effects on collisions efficiency and broadening of SD in a similar set up. In particular, by considering droplets in the range of radii $r <= 70\mu m$, they found that broadening is more important when turbulence is stronger.

Present work, as the authors clearly state, is a sequel of Ref.[1]. Unfortunately, it is much less convincing. As I explain below, I have some major concerns about the results and find the paper lacking a well tought physical analysis.
Here below I report major comments only.

MAJOR COMMENTS:
1) DNS are performed at what is at present considered a low resolution. With $N^3 = 64^3$ grid points, the Eulerian flow is only weakly turbulent. Varying the value of $\epsilon$ does not modify the flow regime from weakly to strongly turbulent (in practice, the Reynolds number stays unchanged), but it impacts all statistics whose prefactors depend on the kinetic energy dissipation rate. In literature, recent studies consider resolutions at $N^3 = 256^3$, at least.
A low resolution set up could be however acceptable if more emphasis were

given to a deep and well-tought analysis of the numerical results. This is not the case of the present paper.

2) In Appendix B, it is stated that droplets dynamics is described by eq. (B10) for $r < 40\mu m$. In this case, the still-fluid terminal velocity is $V_T = g\tau_p = kr^2$. For larger particles, it is unclear if eq. (B10) is still used or not.
How is the droplets dynamics described when $r > 40\mu m$? Is non-linear drag used or what?
Moreover, it is stated that if the radius $r \geq 40\mu m$, the adopted the still-fluid terminal velocity becomes $V_T' = k_2 r$. This means that at $r = 40\mu m$, the function describing $V_T$ not only changes its dependency on $r$, from quadratic to linear, but that there is also a jump in the value: if I am not wrong at $40\mu m$, we have $V_T = 0.19m/s$ and $V_T' = 0.32m/s$....
Either I have not well understod, or there is a problem with this description. Finally, at radii as big as $60 - 100\mu m$, the particle Reynolds number is no longer small, so that I am afraid that the calculation of the disturbance flow in terms of a linear Stokes eq. is no longer valid.
All the big droplets description should be reconsidered and better discussed.

3) The way collisions are treated in the DNS is not described. How are collisions described when one or both droplets have radii larger than 40 micron?

4) A critical issue of this work is the number of simulated droplets, which is initially equal to $80/cm^3$ for a volume of $(10cm)^3$. Since this is not high, I have some troubles with the statistical meaning of the results.
In Figure 1, SD is shown in the range of values $10^2$ down to $10^{-4}$. However below $10^{-3}$, the signal is very noisy, and possibly statistically not relevant. This applies also to all discussion about the size of the largest droplet in the domain: if I have one of such large droplets, its measure is zero. So either the authors are willing to perform many of these simulations to increase the statistical accuracy, or they should limit their discussion e.g. of data in Fig 1. to $dN/dr > 10^{-3}$.

5) Comments in the Results and discussion section are very qualitative. Knowing that "droplets larger than $35\mu m$ (over $0.001cm^{-3}$) can be seen as early as 3.5 minutes in the condensation-collision experiment, but 6 minutes in the collision-only run" might be mentioned, but a physical analysis of the

results is lacking.

Moreover as I said weak and strong turbulence cases differ in the prefactors, not in the amplitude of the inertial range (which is almost absent in DNS at $64^3$), so authors should explore what really causes the observed SD.

Did they measured some conditional statistics to better assess what modifies the droplets collision rates when condensational growth is present? Is there a role of large velocity differences between similar size droplets? I would guess that the so-called sling effect is stronger if r/R approaches 1, and weaker for different size droplets.

6) From literature, including Chen et al. 2016, it is known that turbulence enhancement on collision rate is most significant in similar-sized droplets: what the present work add to this known observation?

7) Also, I think that the purely gravitational case can be omitted.

FINAL ADVICE: I acknowledge that the authors have introduced the "first DNS approach to explicitly study the continuous droplet growth by condensation and collisions inside an adiabatic ascending cloud parcel", but it seems that much of the new physics we can learn of has not been presented here. On the basis of the above considerations, I have to say that in the present form the manuscript is not suitable for publication on ACPL.

**References**

[1] Chen, S., Yau, M. K., and Bartello, P., J. Atmos. Sci., https://doi.org/10.1175/JAS-D-17-0123.1, 2018.

---

## Author Comment (AC1) · 30 Apr 2018

**Reply to comment by W. Grabowski**

We greatly appreciate the reviewer's efforts to carefully review the paper and the insightful comments. We have addressed the questions and concerns indicated in the reviews and believe that the revised version meets the journal's requirements. In the following, blue italic letters denote the reviewer's comments and black regular font letters denote our responses. All the page numbers and line numbers in the response refer to the location in the revision (P=page, L=line)

*1. I found it surprising that the impact of turbulent collisions between similar-sized droplets overwhelms the gravitational collisions between droplets with different sizes. I do not say I do not believe the effect, I just think this should be emphasized more in the manuscript and perhaps supported by additional arguments. Yes, I agree that turbulence dominates the enhancement for equally-sized droplets simply because gravitational collisions vanish in that limit. This is the major problem when showing the turbulent enhancement. However, it is not clear to me why turbulent collisions between equally-sized droplets should outnumber gravitational collisions between droplets of different sizes. The argument presented in the paper, that is, narrowing of the droplet spectra through condensational growth leading to more collisions when turbulence is added, hinges on this conjecture. I think this is the crux of the argument and it should be appropriately stressed in the manuscript. Moreover, one may ask why it is so? Is this because droplets of different sizes tend to cluster in different regions of the turbulent flow (as shown by Lain-Ping Wang in some of his papers), but droplets of the same size should cluster in the same region? Is this the collision efficiency effect? I think it would be appropriate to expand the analysis and maybe come-up with a hand-waving argument to provide some additional support for the key argument.*

The similar-sized collisions outnumbered the different-sized collisions only when both condensational process and strong turbulence are present. By comparing the PDF of droplet collision with and without condensation at different flow conditions (see Fig.3), one can note that the major contributor of similar-sized collisions results from the interaction between the condensational process and the collisional process. When condensation is absent, the number of similar-sized collisions is always small regardless of the intensity of the turbulence (Fig.3, left column). With strong turbulence, the PDF (Fig.3(e)) becomes flattened but the number of different-size collisions remains dominant. This feature ruled out the explanation that the outnumbered similar-sized collisions were purely produced by the clustering effect, the collision efficiency effect, and the transport effect (i.e., by increasing the droplet relative velocity). In contrast, when condensation is present, the intensification of turbulence increases similar-sized collisions (Fig.3, right column or Fig.4, right column), indicating that the enhancement of similar-sized collisions is mainly contributed by the condensation-mediated collisions. In addition, the condensation-mediated collisions only become outnumbered when the turbulence is strong. However, it should also be noted that the large number of similar-sized droplets generated by condensation can further reinforce the clustering effect. Droplet pairs with similar sizes tend to cluster in the same regions of the flow because of similar droplet inertia and terminal velocities. This effect has been confirmed previously in a number of studies

(e.g., Ayala et al. 2008a; Franklin et al. 2005) and is especially pronounced for large droplets. The reason is that small droplets have small Stokes numbers, and they adjust very quickly to changes in the flow and therefore behave more like fluid tracers than inertial droplets.

On the other hand, condensational process tends to narrow the droplet size spectrum and create more similar-sized droplets. We include Figure 5 in the revision (also shown below) to illustrate the time evolution of the pair combinations of droplet with similar sizes (r/R>0.7) and with different sizes (r/R<=0.7). With the presence of condensation, the number of different-sized pairs significantly decreases in the condensation-collision experiment in the first 2 minutes (Fig. 5 (b)). This reduction is caused by the rapid condensational growth for droplets with r <15microns. Simultaneously, the number of similar-sized droplets significantly increases during the first 2 minutes and steadily decreases thereafter (Fig. 5 (d)).  In contrast, when condensation is absent as in the collision-only experiment, the number of different-sized droplet pairs stay relatively constant (Fig. 5 (a)) while the number of similar-sized pairs undergoes only a mild decay (Fig. 5 (c)). The large increase of similar-sized pairs in the collision-condensation experiments during the first 2 minutes significantly increases the number of turbulent-enhanced similar-sized collisions. After 2 minutes, the condensational effect diminishes, and the collision-coalescence process takes over in modulating the droplet pair population. The subsequent decline in the number of similar-sized pairs and the increase in the number of different-sized pairs mainly arise from the collision-coalescence process. We also include a detailed description and explanation of this point in the revision (from P9 L5-P10 L24).

[Figure]

Figure 5: Time evolution of the number of pair combinations for (a) the different-sized droplets (r/R \<= 0.7) and (c) the similar-sized droplets (r/R>0.7) in the collision-only experiments, and (b) the different-size droplets and (d) the similar-sized droplets in condensation-collision experiments. The pair combination is computed using the droplet number concentration (cm$^{-3}$), therefore the unit is cm$^{-6}$. The color denotes the three different flow conditions which are shown in the legend.

*2. I am little concerned with a small size of the computational domain used in the simulations. The size limits the range of scales that the simulations can cover, but this is not what I am worried about. There are some suggestions in the literature that claim the problem also depends on the Reynolds number, that is, the size of the domain, but I feel this is of secondary importance. I am worried about the number of droplets that simulations include. Assuming the total concentration is somewhere around 100 per cc, then you carry around about $10^5$ droplets in your 1 liter domain. To create one 100 micron droplet out of a cloud of droplets with mean size of, say 10 microns, takes of the order of 1000 collisions. Thus, the number of droplets you carry has to significantly decrease with time. Is this a problem? Of course, this also means that you underestimate the impact, correct? I feel one should discuss this issue in the paper (e.g., show how the number of droplet changes with time) and suggest some improvements. One is to run ensemble of simulations to provide confidence intervals on Fig. 1 (the oscillations for radius larger than about 40 microns are a result of a single realization, correct?). The other possibility is to add droplets, for instance, create a new droplet, but keep the colliding droplets in the domain (perhaps re-positioning them randomly). The support for such an approach may come*

*from the following argument: larger droplets fall faster and they simply fall out from the volume you consider and find themselves in the environment that has the same droplet population as before the collision. Such a methodology would provide an upper bound of the impact, correct? I feel it would worthwhile to discuss this aspect in the paper.*

In all the simulations, the total number of collisions remains below 10% of the total number of droplets. Specifically, the number of collisions is below 9% of the total number of droplets at EDR = 500 cm$^2$s$^{-3}$, below 3% at EDR = 50 cm$^2$s$^{-3}$ and below 2% in the purely-gravitational case. Overall, this is a relatively small proportion, and therefore the impact of decreasing droplet number is expected to be secondary. Statistically, the proportion cannot be reduced by either increasing the number concentration or by expanding the domain size. However, it is possible to reduce the statistical uncertainty by doing so. We also agree on the alternative way suggested by the reviewer to reduce the impact of decreasing droplet number by creating new, randomly located droplets after each collision so that the total droplet population is conserved. The justification of this treatment is that larger droplets fall out from the volume and new droplets enter. However, what size of droplets should be introduced remains contentious and needs further justification. We included the above argument and description in the revision (P4, L22-28).

*Minor specific comments (P – page, L – line):*
*1. The text uses the word "observation" and "observe" in several places. I suggest to replace with different words to avoid confusion. This is a numerical not observational study.*

Thanks for the suggestion. We have replaced the words that may lead to confusion in the manuscript.

*2. P2, L26: What you mean by the "large-scale flow" here? Is that the eddy hopping idea as suggested by Grabowski and Wang (the ARFM review) and studied in Grabowski and Abade (JAS 2017)?*

Yes, we have included the references to further clarify the statement.

*3. P4, L15: The 100 micron drop is an arbitrary choice. You can argue it comes from the traditional separation between cloud droplets and drizzle drops, correct? Arguably, one should select a smaller size because the Stokes flow solution that you apply in hydrodynamic interaction calculations is only valid for droplets up to about 50 microns in radius. A comment on that would be desirable here.*

Thanks. Yes, we choose 100 microns as the largest droplet size in accordance of the traditional definition of drizzle drops. We agree that 50 microns is a safe threshold to render the hydrodynamic interaction calculation valid. Droplets larger than 50 microns have particle Reynolds number of order one, which will cause certain inaccuracy of the disturbance flow. However, since the collision efficiency of the droplets over 50 microns is very close to unity due

to the large Stokes number, it is argued that this assumption would not impact much of the collision statistics. We have addressed this issue in the article. (P4, L18-22)

*4. P. 10, L15. I think there should be $R^2$ on lhs of Eq. B1, correct?*

Thanks. We have corrected Eq. B1.

---

## Author Comment (AC2) · 30 Apr 2018

**Reply to comment by Reviewer #2**

We greatly appreciate the reviewer's effort and the helpful comments. We have carefully addressed each question and concern indicated in the comments. Reviewer's comments are in blue italics and our responses are in black regular font. (P=page, L=line)

*MAJOR COMMENTS:*
*1) DNS are performed at what is at present considered a low resolution. With $N^3 = 64^3$ grid points, the Eulerian flow is only weakly turbulent. Varying the value of $\varepsilon$ does not modify the flow regime from weakly to strongly turbulent (in practice, the Reynolds number stays unchanged), but it impacts all statistics whose prefactors depend on the kinetic energy dissipation rate.*
*In literature, recent studies consider resolutions at $N^3 = 256^3$, at least. A low resolution set up could be however acceptable if more emphasis were given to a deep and well-tought analysis of the numerical results. This is not the case of the present paper.*

In Chen et al. (2016), they conducted DNS experiments to investigate the dependency of the collision statistics (i.e., collision rate, relative velocity, and radial distribution function) on the computational Reynolds number (with number of grid points from $64^3$-$1024^3$). It was found that the collision statistics stay constant with the domain size and only increase with the turbulent dissipation rate. The implication is that the collision-related scales are much smaller than the computational domain size, and thus the impact of computational domain sizes is secondary. Similar conclusions can also be found in other DNS studies such as Rosa et al. (2013). Physically, the collision-related scales are comparable to the mean droplet separation distance, which is on a similar order as the Kolmogorov length scale (0.1-1mm) in cumulus clouds. Therefore, we believe $64^3$ is sufficient to reach robust collision statistics. In addition, most of our simulations use $N^3=128^3$ (except for the weak turbulent case with $\varepsilon=50cm^2s^{-3}$ where the domain width already spans 15cm, see Table below). However, we plan to use larger domain sizes for future studies so that more droplets can be included. Since this comment is closely-related to comment (5), we have extended this discussion with the illustration of Fig. A under the second question of comment (5).

Table: domain configuration for the three different flow conditions

| Dissipation rate ($cm^2s^{-3}$) | N | Domain width (cm) |
|---|---|---|
| 0 | 128 | 16.5 |
| 50 | 64 | 14.7 |
| 500 | 128 | 16.5 |

Reference: Chen, S., Bartello, P., Yau, M. K., Vaillancourt, P. A., & Zwijsen, K. (2016). Cloud Droplet Collisions in Turbulent Environment: Collision Statistics and Parameterization. *Journal of the Atmospheric Sciences*, *73*(2), 621-636.

Rosa, B., Parishani, H., Ayala, O., Grabowski, W. W., & Wang, L. P. (2013). Kinematic and dynamic collision statistics of cloud droplets from high-resolution simulations. *New Journal of Physics*, *15*(4), 045032.

*2) In Appendix B, it is stated that droplets dynamics is described by eq. (B10) for r < 40µm. In this case, the still-fluid terminal velocity is $V_T = g\tau_p = kr^2$. For larger particles, it is unclear if eq. (B10) is still used or not.*
*How is the droplets dynamics described when r > 40µm? Is non-linear drag used or what? Moreover, it is stated that if the radius r ≥ 40µm, the adopted the still-fluid terminal velocity becomes $V'_T = k_2r$. This means that at r = 40µm, the function describing $V_T$ not only changes its dependency on r, from quadratic to linear, but that there is also a jump in the value: if I am not wrong at 40µm, we have $V_T = 0.19m/s$ and $V'_T = 0.32m/s$....*
*Either I have not well understod, or there is a problem with this description. Finally, at radii as big as 60—100µm, the particle Reynolds number is no longer small, so that I am afraid that the calculation of the disturbance flow in terms of a linear Stokes eq. is no longer valid.*
*All the big droplets description should be reconsidered and better discussed.*

Equation (B10) is used to describe the forces (drag force and gravity) acted on a droplet. The terminal velocity which determines $\tau_p$ in B10 depends on the size of the droplet, as described in the text after the equation. As for large particles, (B10) is still used but the way a particle responds to the flow has changed from Stokes' drag (as taking the form of $\tau_p \propto r^2$) to an intermediate size (for $40\mu m < r < 0.6mm$, which is described by a linear law $\tau_p \propto r$). Detailed description can be found on P125-126 in Rogers and Yau (1989). It should be noted that the empirical formulas that describe the terminal velocity are developed based on laboratory experiments, and therefore to an extent, bear uncertainty.
The transition of terminal velocity between r<40 µm to r≥40 µm (from $V_T$=0.20 m/s to $V_T$=0.32 m/s) will cause a jump (~37.5% increase) in $\tau_p$, leading to a reduction of the drag force. However, the time step in our simulation is very small (~5*10$^{-5}$ s). Even though this jump modifies the acceleration of the droplet velocity, it only modifies for an infinitesimal time period (one time-step). It follows that the accumulated effect on the change of droplet velocity is negligible. Therefore, we believe that the transition would not impact the statistics in any significant way.

*3) The way collisions are treated in the DNS is not described. How are collisions described when one or both droplets have radii larger than 40 micron?*

The treatment of collision is described in Section 2 (starting from P2 L9). The collisions are treated in the same way for all droplets below 100 microns, i.e., collided droplets merge and form a larger drop. The mass of the newly formed drop is the sum of the two collided droplets, and the central location is the barycenter of the binary system before the collision. For droplets equal or larger than 100 microns, they will be treated as ghost particles, i.e., they neither grow by condensation nor interact/collide with other droplets.

*4) A critical issue of this work is the number of simulated droplets, which is initially equal to 80/cm³ for a volume of (10cm)³. Since this is not high, I have some troubles with the statistical meaning of the results. In Figure 1, SD is shown in the range of values $10^2$ down to $10^{-4}$. However below $10^{-3}$, the signal is very noisy, and possibly statistically not relevant. This applies also to all discussion about the size of the largest droplet in the domain: if I have one of such large droplets, its measure is zero. So either the authors are willing to perform many of these simulations to increase the statistical accuracy, or they should limit their discussion e.g. of data in Fig 1. to dN/dr > $10^{-3}$.*

We agree that for number concentrations below $10^{-3}$ cm$^{-3}$, the droplet number is very small, and the statistics become unreliable. Therefore, the number concentrations below 0.001 cm$^{-3}$ are treated as statistical uncertainty, and all discussion in our result section has excluded those data. We only indicated this in Fig.2 (see caption) in the original version. However, in the revision, we will also include this statement in Fig. 1 as well as the in the main body of the manuscript (P5 L13-14).

*5) Comments in the Results and discussion section are very qualitative. Knowing that \droplets larger than 35μm (over 0.001cm$^{-3}$) can be seen as early as 3.5 minutes in the condensation-collision experiment, but 6 minutes in the collision-only run" might be mentioned, but a physical analysis of the results is lacking.*

Due to the limited number of simulations we conducted, and just a few environmental conditions considered, a conclusion with a definite quantification regarding the exact time of warm rain initiation is difficult. However, we measured the enhancement of the collision frequency in three different flow conditions for different droplet pairs, which gives a quantitative estimation of the turbulence impact on the collisional process. In addition, we provide reasonable physical explanations for the corresponding enhancement by turbulence and the interaction with condensation. The comparison between the results of the condensation-collision experiments and collision-only experiments would be sufficient evidence for the accelerated formation of large droplets.

In a nutshell, the emphases of this paper lie in introducing a sophisticated tool to attack the condensation-collision size-gap challenge and providing preliminary results based on this tool. However, in the long run, this tool will be used to quantify the impact of various microphysical processes on warm rain formation.

*Moreover as I said weak and strong turbulence cases differ in the prefactors, not in the amplitude of the inertial range (which is almost absent in DNS at 64³), so authors should explore what really causes the observed SD.*

We do not agree that the dissipation rate does not change the amplitude of the inertial range. Recall that the spectrum is $\epsilon^{\frac{2}{3}}k^{-\frac{5}{3}}$ in the inertial range and this clearly sets the amplitude of the dissipation range. The Kolmogorov length scale is determined by the dissipation rate: $\left(\frac{v^3}{\epsilon}\right)^{\frac{1}{4}}$.

Therefore, $\epsilon$ in turn modifies the Taylor Reynolds number (as $R_\lambda \propto \left(\frac{L}{\eta}\right)^{\frac{2}{3}}$). More straight-forwardly, one can imagine two turbulent conditions, one with a high dissipation rate and another with a low dissipation rate. The high dissipation rate case has a smaller Kolmogorov length scale according to the above scaling (corresponding to an extension to the right of the tail of the energy spectrum, see $\eta_2$ in Fig. A). By contrast, changing the computational domain size (L, which is in the inertial range) does not actually modify the physical (cloud) Reynolds number. (see Fig. A, changing from L1 to L2).

Concretely, the Reynolds number from DNS only measures the scale separation between the computational domain size and the Kolmogorov length scale and is entirely artificial. This means that modifying the domain size does not guarantee a modification of the flow condition. It is stressed that the dissipation rate is a physical quantity that measures the true intensity of the turbulence, and the computational domain size should not be used as an indicator of turbulence intensity. This is one of the key points of Chen et al. (2016), and further explanation and illustration can be found in their paper.

[Figure]

Figure A: Sketch diagram of energy spectra for two different flow conditions. The spectrum on the top indicates the flow with a higher dissipation rate, and the bottom one indicates a lower dissipation rate. L$_1$ and L$_2$ are the DNS computational domain sizes. $\eta_1$ and $\eta_2$ are the Kolmogorov length scales for the two flow conditions.

*Did they measured some conditional statistics to better assess what modifies the droplets collision rates when condensational growth is present? Is there a role of large velocity differences between similar size droplets? I would guess that the so-called sling effect is stronger if r/R approaches 1, and weaker for different size droplets.*

The mechanism that condensational process modifies the droplet collision rates is described starting from P6 L5. In addition, we have included more supporting evidence (Fig. 5 and its corresponding description from P9 L5-P10 L24) in the revision to further explain the resulting collision statistics.

The role of large relative velocity between similar-sized droplets can be one contributor in modifying the collision rate when condensation is present but not the dominant one. When comparing the collision frequency between collision-only and collision-condensation experiments, one can find that the enhancement mainly exists in similar-sized collisions (see Fig.4 (g-i)). Figure 3 (right column) also demonstrates that when condensation is included, the turbulence impact on similar-sized collisions overwhelmed that of the different-sized collision. However, in collision-only experiment, the increase of the similar-sized collision by turbulence is much smaller than that of the different-sized collision (Fig.3, left column). This implies that that the condensational process creates large number of similar-sized droplets to reinforce similar-sized droplet collisions in turbulence. To further illustrate this mechanism, we also plot the time evolution of pair combination of droplets with similar sizes (r/R>0.7) and with different sizes (r/R<=0.7) (see Fig. 5).

It is obvious that the number of different-sized pairs significantly decreases in the condensation-collision experiment within the first 2 minutes (Fig. 5 (b)) while stays relatively constant in the collision-only experiment (Fig. 5 (a)). Correspondingly, the number of similar-sized droplets in the condensation-collision experiment significantly increases during the first 2 minutes and then steadily decreases thereafter, while this number in the collision-only experiment only sees a mild decay. The substantial increase of similar-sized pairs in the collision-condensation experiments during the first 2 minutes significantly increases the turbulent-enhanced similar-sized collisions. The condensational effect diminishes after 2 minutes, and the collision-coalescence process takes over in modulating the droplet pair population. The subsequent decline in the similar-sized pairs and the increase in the different-sized pairs are mainly caused by the collision-coalescence process.

[Figure]

Figure 5: Time evolution of the number of pair combinations for (a) the different-sized droplets (r/R \<= 0.7) and (c) the similar-sized droplets (r/R>0.7) in the collision-only experiments, and (b) the different-size droplets and (d) the similar-sized droplets in the condensation-collision experiments. The pair combination is computed using the droplet number concentration (cm$^{-3}$), therefore the unit is cm$^{-6}$. The color denotes the three different flow conditions which are shown in the legend.

*6) From literature, including Chen et al. 2016, it is known that turbulence enhancement on collision rate is most significant in similar-sized droplets: what the present work add to this known observation?*

As mentioned previously, the main purpose of this study is to introduce a realistic and accurate tool to study the condensation-collision size gap problem. Past DNS studies often neglect either condensation or collision, and thus the possible contribution to the DSD broadening by the interactions between the two growth processes is omitted. This paper provides one of the first DNS models that simulate continuous, seamless droplet growth by condensation and collision.

In conclusion, this study demonstrates that the inclusion of condensation significantly increases the turbulence enhancement on droplet collisions and provides one of the first direct evidence and interpretation of how turbulence impacts the interaction of condensation and collision processes.

*7) Also, I think that the purely gravitational case can be omitted.*

The purely gravitational case is provided as a controlled experiment in this study. It serves as a comparison to the turbulent cases to demonstrate the impact of turbulence on collisions and on condensation-induced collisions. Therefore, it is important to display the purely-gravitational case in the paper.

*FINAL ADVICE: I acknowledge that the authors have introduced the "first DNS approach to explicitly study the continuous droplet growth by condensation and collisions inside an adiabatic ascending cloud parcel", but it seems that much of the new physics we can learn of has not been presented here. On the basis of the above considerations, I have to say that in the present form the manuscript is not suitable for publication on ACPL.*

It is not very clear what "new physics" are referred to here. However, this paper provides a novel solution to the condensation-collision size-gap problem. The size-gap problem has puzzled cloud physicists for decades. The new explanation on turbulence-enhanced rain formation through the interaction of condensation and collision is the new physics which has tremendous value for the cloud physics community.